# Type 1 Sodium Calcium Exchanger Forms a Complex with Carbonic Anhydrase IX and Via Reverse Mode Activity Contributes to pH Control in Hypoxic Tumors

**DOI:** 10.3390/cancers11081139

**Published:** 2019-08-09

**Authors:** Veronika Liskova, Sona Hudecova, Lubomira Lencesova, Filippo Iuliano, Marta Sirova, Karol Ondrias, Silvia Pastorekova, Olga Krizanova

**Affiliations:** 1Institute of Clinical and Translational Research, Biomedical Research Center, Slovak Academy of Sciences, Dúbravská cesta, 84505 Bratislava, Slovakia; 2Institute of Virology, Biomedical Research Center, Slovak Academy of Sciences, Dúbravská cesta, 84505 Bratislava, Slovakia; 3Department of Chemistry, Faculty of Natural Sciences, University of Ss. Cyril and Methodius, Námestie J. Herdu 2, 91701 Trnava, Slovakia

**Keywords:** sodium calcium exchanger type 1, sodium proton exchanger type 1, carbonic anhydrase IX, hypoxic tumors, intracellular pH

## Abstract

Hypoxia and acidosis are among the key microenvironmental factors that contribute to cancer progression. We have explored a possibility that the type 1Na^+^/Ca^2+^ exchanger (NCX1) is involved in pH control in hypoxic tumors. We focused on changes in intracellular pH, co-localization of NCX1, carbonic anhydrase IX (CA IX), and sodium proton exchanger type 1 (NHE1) by proximity ligation assay, immunoprecipitation, spheroid formation assay and migration of cells due to treatment with KB-R7943, a selective inhibitor of the reverse-mode NCX1. In cancer cells exposed to hypoxia, reverse-mode NCX1 forms a membrane complex primarily with CA IX and also with NHE1. NCX1/CA IX/NHE1 assembly operates as a metabolon with a potent ability to extrude protons to the extracellular space and thereby facilitate acidosis. KB-R7943 prevents formation of this metabolon and reduces cell migration. Thus, we have shown that in hypoxic cancer cells, NCX1 operates in a reverse mode and participates in pH regulation in hypoxic tumors via cooperation with CAIX and NHE1.

## 1. Introduction

Tumor hypoxia is a constantly evolving factor of tumor tissue microenvironment that critically affects tumor growth and fate. Clinically, hypoxia is associated with poor patient prognosis and resistance to chemotherapy and radiotherapy. Development of new strategies targeting hypoxic tumors is critical for improving patients’ outcome. Moreover, hypoxia is closely associated with an interstitial acidosis [1], which facilitates metastatic dissemination of cancer cells [2].

Majority of solid tumors exhibit acidic extracellular pH (pHe) and slightly alkaline intracellular pH (pHi). Such reversal of the pH gradient is detectable even in the preliminary stages of tumorigenesis and is crucial for survival and expansion of tumors, regardless of their pathology, genetics and origins [3]. Moreover, this phenomenon seems to be an ubiquitous feature of all malignant tumors. Most of the research in this clinically important area has focused on proton extrusion, in particular via the Na^+^/H^+^ exchanger 1 (SLC9A1, NHE1) and various H^+^/ATPases. Also, HCO_3_^−^ transporters such as the electroneutral Na^+^, HCO_3_^−^ cotransporter (SLC4A7, NBCn1), are upregulated and play central roles in pH regulation by mediating import of HCO_3_^−^ ions [4].

Hypoxic tumors are characterized by the expression of carbonic anhydrase IX (CA IX), a regulator of pH and tumor growth [5]. CA IX is a hypoxia-induced catalytic component of the HCO_3_^−^ import arm of the pH control machinery [6]. Catalytic domain of this enzyme is exposed at the surface of cells and contributes to acidification of the outer microenvironment, since it produces extracellular protons and accelerates CO_2_ diffusion. Also, it promotes neutralization of intracellular pH by facilitating HCO_3_^−^ uptake and lactate export [7]. Catalytic domain/activity of CA IX was shown to play a role in cell migration [8]. Importantly, acidification of the tumor microenvironment stimulates breakdown of the extracellular matrix, and promotes migration, invasion and metastasis [9,10]. Disrupting pH homeostasis by blocking CA IX with specific antibodies and/or selective CA IX inibitors might broadly relieve the common resistance of hypoxic tumors to anticancer therapy [11,12].

Sodium hydrogen exchanger (NHE) is a potent acid-extruding membrane transport protein. Upregulation of the expression and/or activity of one of nine NHE isoforms-NHE1, commonly correlates with tumor malignancy. NHEs transport one H^+^ out of cells in exchange with one Na^+^ into cells. Thus, NHE1 is an important regulator of both pHi and pHe in tumors [13]. Chronic hypoxic exposure causes an increased NHE activity and H^+^ efflux. NHE1 activity (to maintain resting pHi) is fundamental for growth in hypoxia [14,15]. During intracellular acidification, allosteric modification at the proton binding site leads to stimulation of NHE1 activity, that can be modulated through the C-terminal tail region. These regulatory proteins include carbonic anhydrase II, calmodulin and calcineurin homologous protein 1 and 2, actin-binding ERM proteins, heat shock protein 70, etc. [16].

Sodium/calcium exchanger (NCX) is definitely an important player in the regulation of intra-cellular calcium homeostasis. Three types of the NCX were found and characterized up to now: type 1 (which is ubiquitous), type 2 (which was found to be expressed in brain) and type 3 (expressed in brain and skeletal muscle) [17,18]. However, type 1 NCX (NCX1) is the most abundant and therefore best studied type. Although majority of papers dealing with the NCX1 were performed on the cardiac tissue [19,20], importance of this calcium transporter was documented also in other types of cells and tissues, e.g., brain [21,22], endocrine system and immune system [23], etc., where it can be involved in apoptosis and proliferation [24]. Also, NCX1 levels are up-regulated by hypoxia [20,25]. Although the forward mode of NCX represents a major physiological form, the reverse mode of NCX becomes predominant in pathological settings [26]. Coupling of the reverse mode NCX with specific ion transporters may mediate net fluxes of Na^+^ and/or Ca^2+^, which may effectively integrate and regulate important physiological events on the cellular and systemic levels [27,28]. Hypoxia-induced Ca^2+^ increase in smooth muscle cells of fetal small pulmonary arteries results from cytosolic Ca^2+^ influx mediated primarily by the reverse mode of the NCX [29]. Also, calcium influx via the reverse mode NCX is involved in the cascade of NO-induced neuronal apoptosis and NO activation of the NCX through the guanylate cyclase/PKG pathway. Sodium nitroprusside, a NO donor, causes Ca^2+^ overload, which may be mediated by the reverse mode of the NCX action. Consequently, the NCX-mediated Ca^2+^ influx activates a Ca^2+^-dependent cell death machinery, leading to apoptotic-like cell death [30]. Recently, it was shown that KB-R7943, a selective inhibitor of the reverse mode NCX1, promotes cell death in prostate cancer cells by activating the JNK signaling pathway and blocking autophagic flux [31]. Moreover, reverse mode NCX1 was shown to operate in metastatic human melanoma cells [32]. Recently, we have shown that in DLD1 and A2780 cells, NHE1 and NCX1 form complexes that can be partially disrupted due to treatment with a slow sulfide donor GYY4137 leading to the intracellular acidification. H_2_S-induced disruption of NHE1/NCX1 complex is associated with overexpression of these proteins and internalization of NHE1 [33].

Up to now, little is known about the role of sodium/calcium exchanger in development and progression of tumor growth. Also, to our knowledge, cooperation of the CA IX and NHE1 has not been described. Thus, the goal of the present work was to elucidate a role of the type 1 NCX in pH regulation in hypoxic tumors and to examine its cooperation with CA IX and NHE1.

## 2. Results

### 2.1. Changes in Intracellular pH in SiHa, DLD1 and A2780 Cells in Response to KB-R7943 Treatment

During exposure of SiHa, DLD1, and A2780 cells to 1% hypoxia, intracellular pH underwent considerable changes as shown on Figure 1A–C. After initial 24 h of incubation, a significant intracellular acidification could be observed in all these cells. Later on, after 48 h of hypoxia, the intracellular pH became slightly alkaline. Treatment of cells with KB-R7943, a specific blocker of the NCX1 reverse mode, also led to acidification of intracellular pH under normoxic conditions with no difference in pHi between 24 and 48 h incubation periods (KBR, Figure 1A–C). When KB-R7943 was added to hypoxic cells (HyKBR) for 48 h, intracellular pH remained acidic and did not alkalize in contrast to untreated hypoxic cells (Figure 1A–C). These data suggest that NCX1 operating in the reverse mode, contributes to control of intracellular pH in hypoxic cancer cells. Involvement of the NCX1 in the pHi regulation in hypoxia was verified by NCX1 knock-down using NCX1 siRNA, where clear drop of the intracellular pH was observed after 48 h of hypoxia compared to scr siRNA (Figure 1D).

### 2.2. Co-localization of the NCX1 and CA IX Determined by Proximity Ligation Assay (PLA) in SiHa and RCC4 Cells

Further, we evaluated a co-localization of NCX1 with CA IX, an established hypoxia-induced pH regulator, by confocal microscopy. We observed clearly overlapping signals in SiHa cells after 48 h of exposure to 1% hypoxia (Figure 2A). Moreover, proximity ligation assay revealed a distinct red signal supporting the view that NCX1 co-localizes and potentially interacts with CA IX in hypoxic SiHa cells (Figure 2B). This signal was partially diminished upon NCX1 silencing by NCX1 siRNA (Figure 2B), since the effectivity of NCX1 silencing was approximately 50%. Blocking of the reverse mode NCX1 by addition of KB-R7943 to hypoxic cells for 48 h completely prevented the PLA detectable co-localization signal (HyKBR). This observation suggests that either both proteins are not in a close proximity, or KB-R7943 forms a sterical barrier between these proteins. In RCC4 cells that are pseudo-hypoxic due to VHL mutation and constitutive upregulation of the hypoxia-induced molecular pathways, red signal showing co-localization of the CA IX and NCX1 was clearly visible in control cells (No; Figure 2C) and was significantly decreased in cells treated with KB-R7943 (KBR; Figure 2C).

### 2.3. Formation of NCX1/CA IX/NHE1 Metabolon and Determination of the Cytosolic Ca^2+^ and Na^+^ Levels in Hypoxic SiHa, DLD1 and A2780 Cells

Based on the evidence that NCX1 participates in regulation of intracellular pH, we wanted to confirm that NCX1 cooperates with both CA IX and NHE1. Therefore, we performed a series of immunoprecipitations (Figure 3). After 48 h of 1% hypoxia, NCX1 co-precipitated with CA IX as well as with NHE1. The corresponding profile was obtained by using CA IX-mediated co-precipitation (Figure 3A), thus suggesting a formation of a metabolon, which is potentially involved in pH regulation. As demonstrated by the above-mentioned results (Figure 1 and Figure 2), in this metabolon NCX1 operates in the reverse mode. We proved this orientation of the NCX1 by determining cytosolic Ca^2+^ and Na^+^ levels in SiHa, DLD1 and A2780 cells (Figure 3B,C). In normoxia, KB-R7943 did not change significantly a level of cytosolic Ca^2+^ compared to normoxia, as shown in our recent paper [33]. Cytosolic Ca^2+^ was markedly increased after the exposure to hypoxia and this increase was prevented by adding KB-R7943 (Figure 3B). In accordance with that, decrease in cytosolic Na^+^ due to hypoxic conditions observed after 24 h was prevented by KB-R7943 (Figure 3C). However, after 48 h of hypoxia, levels of cytosolic Na^+^ did not show any further decrease, which might suggest involvement of additional Na^+^ transporter (possibly NHE1) in pH handling (Figure 3C).

After 48 h, NHE1 is bound to the NCX1, as determined by immunoprecipitation (Figure 4A). Again, KB-R7943 decreased the signal of NHE1 co-precipitated with NCX1 (Figure 4A). In SiHa cells, silencing of the NCX1 revealed similar decrease in the NHE1 (Figure 4A). Proximity ligation assay performed on RCC4 cells verified co-localization of the NCX1 and NHE1 (No; Figure 4B), which was significantly reduced in the presence of KB-R7943 (KBR; Figure 4B). Negative control (NC), where NCX1 antibody was omitted, proves the specificity of the PLA signal.

### 2.4. Role of the NCX1 in Spheroid Formation

Importance of the NCX1 functioning in the reverse mode was also supported by spheroid formation assay (Figure 5).

When spheroids were grown in the presence of KB-R7943, their disruption occurred much earlier compared to non-treated spheroids (Figure 5A,C). Amount of CA IX, NCX1 and HIF-1α proteins in spheroids was determined at the end of the experiment and showed no difference in the absence versus presence of KBR (Figure 5B). Percentage of apoptotic (striped columns) and necrotic (black columns) cells in spheroids was determined on days 8 and 10 (Figure 5D). We observed a significant increase in a number of necrotic cells upon KB-R7943 treatment, when compared to untreated spheroids.

### 2.5. Effect of the Reverse Mode NCX1 on Cell Migration

Since pH regulation is known to affect migration capacity of cancer cells, the proposed pH modulating role of the reverse mode of the NCX1 was evaluated by the migration assay. KB-R7943 significantly decreased the migration of SiHa cells (Figure 6A) and also SiHa cells treated with dimethyloxalylglycine (DMOG; Figure 6B), a cell permeable prolyl-4-hydroxylase inhibitor, which upregulates hypoxia-inducible factor (HIF). In RCC4 cells, area of the wound closure after 36 h was also significantly broader in cells treated with KB-R7943 than in untreated cells (Figure 6C), supporting the view that the reverse mode of the NCX1 is implicated in cell migration under hypoxia.

## 3. Discussion

Regulation of intracellular and extracellular pH in hypoxic tumors significantly affects the fate of tumor cells and therefore, understading its mechanisms and components is of crucial relevance for cancer biology. In this study we showed that the NCX1 operating in the reverse mode contributes to pH regulation in hypoxic tumors. We demonstrated that silencing of the NCX1 gene causes decrease of intracellular pH in hypoxic SiHa cells and that KB-R7943, a specific blocker of the reverse mode of NCX1 diminishes the abilty of SiHa, DLD1 and A2780 cancer cells exposed for 48 h to 1% hypoxia to control their intracellular pH. These observations are in agreement with our previous findings in DLD1 and A2780 cells [33]. Based on results of this study we propose the formation of a metabolon (Figure 7), composed of NCX1, NHE1 and CA IX under hypoxic conditions. It was already shown that in certain cancer cells, the NCX1 operates in the reverse mode that allows transport of Ca^2+^ ions into the cell, while extruding sodium ions out of the cell [31,32]. Resulting Ca^2+^ overload of cancer cells was suggested to be responsible for tumor calcification [34], which in turn is linked with the presence of necrosis as an extreme manifestation of hypoxia. Interestingly, higher expression levels of hypoxia-induced CA IX tended to be associated with nonlinear calcification in breast tumors [35]. Similarly, changes in local pH regulation appear to affect the extent and pattern of calcification in tumors and thus, it is conceivable that CA IX can contribute to this process. Our data suggest that it can do it also via cooperation with NCX1 functioning in the reverse mode.

Indeed, it is known that calcification is significantly more common in metastatic than in nonmetastatic lymph nodes [36], which might be also caused by the reversed action of the NCX1. Intracellular acidification induced by metabolic changes associated with hypoxia mobilizes mechanisms that can extrude proton ions from cells. Overload of proton ions in the cells together with relatively low amount of Na^+^ ions due to the action of reversed NCX1 can activate the hypoxia-inducible NHE1, as it was already shown in our previous work [33]. This process can involve a pH sensing site on the internal/cytoplasmic surface of the NHE1 that, if protonated, activates the transport [10]. NHE1 helps to maintain an alkaline intracellular pH by transporting protons out of the cell against Na^+^ ions that are transported into the cell [37]. These Na^+^ ions can feed the reverse mode of NCX1 in exchange for Ca^2+^ uptake. Moreover, this action can be further potentiated by hypoxia-induced CA IX, that catalyzes the reversible hydration of CO_2_ to bicarbonate ions HCO_3_^−^ and protons (H^+^) at the extracellular surface and via interaction with Na^+^ HCO_3_^−^ transporter (NBC1) facilitates import of HCO_3_^−^ ions [6]. Thereby, CA IX controls intracellular and extracellular acid-base balance that regulates both survival and invasive properties. Recently, it was shown that disrupting pH homeostasis by blocking HCO_3_^−^ import might broadly relieve the common resistance of hypoxic tumors to anticancer therapy [11]. Thus, we speculate that NBC1 might be a part of this metabolon (see the Figure 7). NBC1 is one of the major alkalinizing mechanisms in the cardiomyocytes [38]. Since both NHE1 and NBC1 increase intracelular Na+ concentration, the reverse mode NCX1 may serve as a major transport system extruding Na^+^ from the intracellular space. Based on our results, we hypothesized about the physiological relevance of this complex. Since CA IX as well as NHE1 are activated in hypoxic conditions and participate in proton extrusion, proton transport from cells could be considered as an adaptive mechanism in hypoxia that is essential for the cell’s survival [37,39]. Although functions of these proteins in hypoxic conditions are known, their mutual co-operation and co-localization with NCX1 was not described until now. We have shown that KB-R943 as a blocker of the NCX1 reverse mode reduced migration rate of RCC4 cells. Role of the NCX migration was already described, e.g., in human gastric myofibroblasts [40] or microglia [41]. Complex formation probably increases extracellular pH in a microdomain, which might boost migration specifically from that region. On this basis, we propose that the complex assembled from NCX1, CA IX and NHE1 participates in intracellular alkalinization and extracellular acidification thereby creating a local microenvironment, which might boost cell migration. Because hypoxia and acidosis are linked with aggressive tumor phenotype, identification of the metabolon described in this study opens new opportunities for more efficient targeting of the its components. This can be accomplished either individually or in combination using specific antibodies and/or selective inhibitors that interfere with the related adaptive pathways and show anticancer effects, as described for example for CA IX inhibitors [12].

## 4. Materials and Methods

### 4.1. Cell Cultivation and Treatment

For experiments, human cervical carcinoma cell line SiHa (ATCC^®^ HTB-35^TM^), renal cell carcinoma cell line RCC4 (ECACC, 03112702), ovarian cancer cell line A2780 (Sigma-Aldrich, 93112519) and/or colon adenocarcinoma cell line DLD1 (ATCC, CCL-221), were cultured in Dulbecco’s Minimal Essential Medium (DMEM; Sigma, St. Louis, MO, USA) or RPMI medium (Sigma) with a high glucose (4.5 g/L) and L-glutamine (300 μg/mL), supplemented with 10% fetal bovine serum (Sigma), penicillin (Calbiochem, San Diego, CA, USA; 100 U/mL) and streptomycin (Calbiochem; 100 μg/mL). Cells were cultured in a water-saturated atmosphere at 37 °C and 5 % CO_2_. After plating, cells were treated for 24 or 48 h with KB-R7943 mesylate (Santa Cruz Biotechnology, Dallas, TX, USA, 10 µmol/L). For some experiments, cells were grown for 24 or 48 h in hypoxic chamber (1% O_2_, 2%H_2_, 5%CO_2_, (91%N_2_; Baker Ruskinn Technology, Maastricht, The Netherlands) or treated with dimethyloxalylglycine (DMOG, 100 μM). Following numbers of cells were used for pH determination and also for determination of cytosolic Ca^2+^ and Na^+^: SiHa—1.5× 10^5^; DLD1—2× 10^5^; A2780—2× 10^5^.

### 4.2. Measurement of Intracellular pH

To determine changes in the intracellular pH (pHi), pH sensitive cell permeabile fluorescent probe 2′,7′-biscarboxyethyl-5,6-carboxyfluorescein-acetoxymethyl ester (BCECF-AM; Sigma Aldrich) was used as described in Pastorek et al. [5]. Briefly, cells were loaded with 10 µmol/L BCECF and 0.5% pluronate in cultivation media without FBS for 45 min at 37 °C, 5 % CO_2_, in dark. After subsequent washing with PBS buffer, fluorescence was measured at 490/535 nm and 440/535 nm on Synergy fluorescence scanner (BioTek, Bad Friedrichshall, Germany). The pHi signal was calibrated to bvpH0 by adding 10 µmol/L nigericin (Sigma Aldrich) with 130 mmol/L KCl. ΔpHi was ratiometric calculated from values obtained at 490 and 440/535 nm.

### 4.3. Immunofluorescence

Cells grown on glass cover slips were fixed in ice-cold methanol for 5 min. Non-specific binding was blocked by incubation with PBS containing 3% BSA for 1 h at 37 °C. Cells were then incubated with primary antibodies for 1 h at 37 °C and washed four times with PBS containing 0.02% Tween 20 for 10 min. The following primary antibodies were used: anti-human CA IX mouse monoclonal antibody M75 in undiluted hybridoma medium [5], rabbit polyclonal antibody NCX1 (p11-13, Swant, Bellizona, Switzerland) diluted 1:200 in PBS with 1% BSA. After washing the cells were inclubated with Alexa-conjugated secondary antibodies (Alexa Fluor 594 goat anti-mouse IgG and Alexa Fluor 488 donkey anti-rabbit IgG) diluted 1:1000 in PBS-BSA for 1 h at 37 °C, and washed four times with PBS. Finally, cells were mounted onto slides in mounting medium with Citifluor (Agar Scientific Ltd., Essex, UK) analyzed by an LSM 510 Meta confocal microscope (Zeiss, Jena, Germany). Images were taken with Plan Neofluar 40×/1.3 oil objective at optical zoom 2 in multi-track mode. The acquired images were processed in ImageJ, the Co-localization. Finder plug-in was used to assess the co-localization of CA IX and NCX1 signals. The background pixels were excluded from the co-localization analysis, the co-localizing pixels were determined and marked in green.

### 4.4. Proximity Ligation Assay

The proximity ligation assay (PLA) was used for in situ detection of the interaction between CA IX/NCX1 and NCX1/NHE1. The assay was performed in a humid chamber at 37 °C according to the manufacturer’s instructions (Olink Bioscience, Uppsala, Sweden). SiHa cells were seeded on glass coverslips and allowed to attach before transfer to 1% hypoxia and further cultured for 48 h. RCC4 cells were seeded on glass coverslips and not exposed to hypoxia, since these cells are pseudohypoxic due to VHL mutation. The cells were fixed with methanol, blocked with 3% BSA/PBS for 30 min, incubated with a mixture of antibodies against CA IX and NCX1 or NCX1 and NHE1 for 1 h, washed three times, and incubated with plus and minus PLA probes for 1 h. Then, the cells were washed (3 × 5 min), incubated for 40 min with ligation mixture containing connector oligonucleotides, washed again, and incubated with amplification mixture containing fluorescently labeled DNA probe for 100 min. After a final wash, the samples were mounted and the signal representing interaction between CA IX and NCX1 or NCX1 and NHE1 was analyzed using a Zeiss LSM 510 Meta confocal microscope with a Plan Neofluar 40×/1.3 oil objective. The following antibodies were used: rabbit polyclonal antibody NCX1 (π11-13, Swant), mouse monoclonal CA IX antibody M75 in undiluted hybridoma medium [5]. and mouse monoclonal NHE1 (SLC9A1, Sigma-Aldrich).

### 4.5. Immunoprecipitation

Appropriate monoclonal antibody was incubated with 60 µL washed magnetic beads (Dynabeads M-280), coated with M-280 sheep anti-mouse IgG (Invitrogen Dynal AS, Oslo, Norway) for overnight at 4 °C on a rotator (VWR International, Radnor, PA, USA). As negative controls, the coated beads were incubated with either mouse IgG1κ (MOPC21, Sigma) for mAb, or with rabbit γ-globulin (Jackson ImmunoResearch, West Grove, PA, USA) for pAb raised in rabbits. The beads with attached antibody were washed (twice, 200 μL) with phosphate-buffered saline (PBS). Proteins were immunoprecipitated from 1 mg of detergent-extracted total protein by incubation for 4 h at 4 °C with antibody-bound beads. Bead complexes were washed with (four times 200 μL) PTA solution (145 mmol/L NaCl, 10 mmol/L NaH_2_PO_4_, 10 mmol/L sodium azide, and 0.5% Tween 20, pH 7.0). Immunoprecipitated proteins were then extracted with 60 μL of 2× Laemmli sample buffer (Bio-Rad, Hercules, CA, USA) and boiled for 5 min. Following antibodies were used for immunoprecipitation: mouse monoclonal antibody to NCX1 (R3F1, Swant), rabbit polyclonal antibody to NCX1 (p11-13, Swant), mouse monoclonal antibody M75 to CA IX [5].

### 4.6. Western Blot Analysis

Firstly, cells were scraped into 10 mmol/L Tris-HCl, pH 7.5, 1 mmol/L phenylmethyl- sulfonylfluoride (PMSF, Serva, Heidelberg, Germany), protease inhibitor cocktail tablets (Complete EDTA-free, Roche Diagnostics, Mannheim, Germany) and centrifuged for 10 min at 3000× *g* at 4 °C. The pellet was re-suspended in Tris-buffer containing the 50 µmol/L CHAPS (3-[(3-cholamidopropyl) dimethylammonio] 1-propanesulfonate, Sigma), and then incubated for 20 min at 4 °C. Concentration of proteins in the lysate was determined by the method of Lowry et al. [42]. 20–60 µg of protein extract from each sample was loaded on 8–16% and 4–12% SDS polyacrylamide gradient gels (Amersham Biosciences, Amersham, Buckinghamshire, UK). Afterwards, proteins were transferred to the Hybond-PVDF membrane (Amersham Biosciences) using semidry blotting (Owl Inc., Thermo Scientific, Waltham, MA, USA). Membranes were blocked in 5% non-fat dry milk in Tris-Buffered Saline with Tween 20 (TBS-T) overnight at 4 °C and then incubated for 1 h with appropriate primary antibody. Following washing, membranes were incubated with secondary antibodies to mouse or rabbit IgG conjugated to horseradish peroxidase for 1 h at room temperature. An enhanced chemiluminiscence detection system (Luminata Crescendo Western HRP Substrate, Millipore, Burlington, MA, USA) was used to detect bound antibody. The optical density of individual bands was measured on Kodak Image 440 device (Kodak, Zaventem, Belgium) and quantified using PCBAS 2.0 software.

Antibodies raised against the following proteins were used: rabbit polyclonal antibody to NCX1 (p11–13 Swant), mouse monoclonal antibody to NCX1 (C2C12 Abcam, Cambridge, UK), rabbit polyclonal antibody to NHE1 (Abcam), mouse monoclonal antibody to CA IX [5].

### 4.7. Cytosolic [Ca^2+^]_i_ Staining by Fluo-3AM Fluorescent Dye

Cells were plated on a 24-well plate at the density described in the Section 4.1. After the treatment, cells were washed with 1 mL of serum-free medium and loaded with 2 µM Fluo-3AM; (Sigma-Aldrich, USA) in the presence of 0.5% pluronate (Sigma-Aldrich) for 60 min at 37 °C, 5% CO_2_ in the dark. Cells were then washed three times with a 500 µL of serum-free medium. Fluorescence was measured on Synergy H1 Hybrid Multi-Mode Reader (BioTek, Bad Friedrichshall, Germany) at λ_ex_ 506 nm; λ_em_ 525 nm. Afterwards, 1 mM CaCl_2_ and 10 μM ionomycin were added to each well and fluorescence was measured again. Results were expressed as the arbitrary units of fluorescence.

### 4.8. Cytosolic [Na^+^]_i_ Staining by SBFI-AM Fluorescent Dye

Intracellular concentration of sodium [Na^+^]_i_ was measured according to Sathish and co-workers [43] with small modifications. After loading with 5 µmol/L SBFI-AM in the presence of 0.1% pluronate, 10 µmol/L gramicidin and 100 µmol/L ouabain in the serum free DMEM for 3 h in CO_2_ incubator (hypoxic cells in 1% hypoxia), cells were washed twice with 5% glucose. For measurement, excitation 340/380 nm and emission 500 nm was used. Results are displayed as ratio of these two values. Calibration curve was made from mixture of two solutions with equal ionic strength (Na^+^ and potassium gluconate) with different concentrations of Na^+^ and K^+^ ions (0, 5, 20, 50, 70, 100, 145 mmol/L).

### 4.9. NCX1 Silencing by siRNAs

Cells were grown in 6-well plates in DMEM with 10% FBS to the density of 1 × 10^5^. Transfection of siRNAs was performed with DharmaFECT1 (Dharmacon, Thermo Scientific, Lafayette, CO, USA) according to producer’s protocol. ON-TARGET plus mixture of siRNAs for human SLC81 (NCX1; Dharmacon, Thermo Scientific) was applied to the final concentration of 50 pmoL per well. The same procedure was performed with Non-Targeting plus siRNAs for scrambled control. Hypoxic samples were immediately placed into the hypoxic chamber for 24 or 48 h. After this period, cells were harvested and used in further experiments. Efficacy of the NCX1 silencing was verified by western blot analysis and was approximately 50%.

### 4.10. Spheroids

SiHa spheroids were generated by seeding 5000 cells/well in 96 well plate previously coated with 1% agarose. After 4 days of incubation at 37 °C and 5% CO_2_ the treatment began by adding fresh medium and diluted compounds in DMEM. In all the experiments 30 replica wells were set up for control and for each compound. Medium was changed every 48 h and pictures (by Axiovert 40 CFL microscope, Zeiss) were taken to check integrity and measure the spheroids diameter. In a parallel experiment CA IX expression has been evaluated at the onset of the treatment by Western blot. After 8 and 10 days of treatment cell viability has been evaluated with propidium iodide by Guava EasyCyte 6HT flow cytometer (Millipore, Burlington, MA, USA). Briefly, spheroids were collected, rinsed with PBS and disrupted by trypsin/EDTA. After been counted, the cells were resuspended in a 5 µg/mL propidium iodide (PI) solution, kept on ice for 5 min and analysed by flow cytometer.

### 4.11. Detection of Apoptosis with Annexin-V-FLUOS

Spheroids were pelleted at 1000 rpm for 5 min. Cells were then washed with 1ml of PBS, cell pellet was resuspended in 200 μL of Annexin-V-FLUOS/ propidium iodide labeling solution (Roche Diagnostics, Indianapolis, IN, USA) and incubated at room temperature in dark for 20 min according to the manufacturer´s protocol. After the incubation, samples were diluted with 400 µL PBS, placed on ice and measured on BD FACSCanto II flow cytometer (Becton Dickinson, Ann Arbor, MI, USA). Results were evaluated with a Flowing software version 2.5.1.

### 4.12. Cell Migration Assay

Fifteen thousand SIHA cells and 18000 RCC4 cells per well were plated on ImageLock 96-well plates (Essen BioScience, Ann Arbor, MI, USA), and let to adhere for 24 h. Confluent monolayers were then wounded with wound making tool (IncuCyte WoundMaker; Essen BioScience), washed twice and supplemented with fresh culture medium. Some groups of cells were treated with KB-R7943 (10 μmol/L). Images were taken every 2 h for the next 36 h in the IncuCyte ZOOM™ kinetic imaging system (Essen BioScience). Cell migration was evaluated by IncuCyte ZOOM™ 2016A software (Essen BioScience) based on the relative wound density measurements and expressed as means of octaplicates ± SEM.

### 4.13. Statistical Analysis

Each value represents an average of 3–6 wells from at least two independent cultivations of SiHa, RCC4, DLD1 and A2780 cells. The results are presented as the mean ± S.E.M. Significant differences between the groups were determined by one-way ANOVA. For multiple comparisons, an adjusted t test with *p* values corrected by the Bonferroni method was used. Statistical significance of migration was evaluated by nonparametric Wilcoxon-Mann-Whitney test and Kruskal-Wallis nonparametric test.

## 5. Conclusions

In summary, in the present study we show that the reverse-mode NCX1 forms a membrane complex with the CA IX and NHE1. This membrane complex not only represents a potent modulator of intracellular/extracellular pH, but can play a role in tumor cell´s migration (and potentially metastasis formation). Since NCX1 also forms complexes with the β1- and β3- but not β2-adrenoceptors [44], it is likely that this metabolon is assembled of additional protein components that can further modulate cellular responses to hypoxia and pH imbalances in tumor microenvironment.

## Figures and Tables

**Figure 1 cancers-11-01139-f001:**
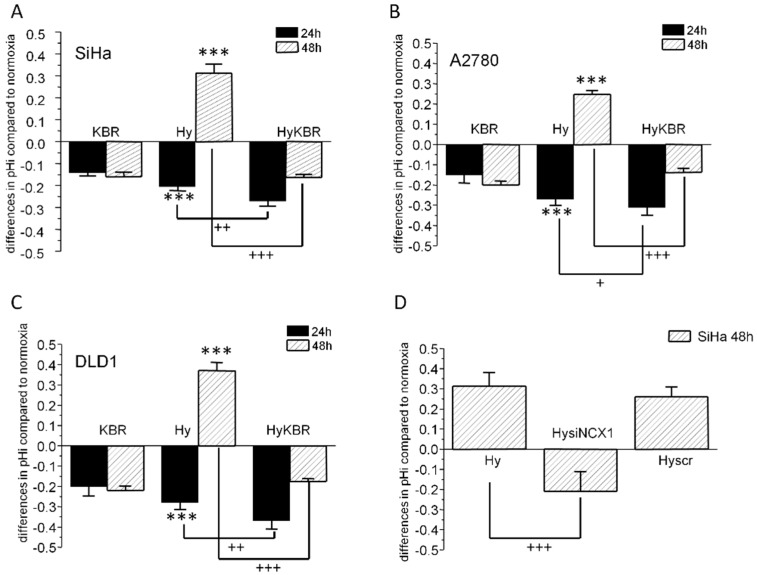
Role of the NCX1 in intracellular pH regulation. Firstly, we determined an effect of the reverse mode NCX1 blocker KB-R7943 on intracellular pH in SiHa (**A**), A2780 (**B**) and DLD1 (**C**) cells incubated for 24 h (black columns) and 48 h (striped columns) in hypoxia. Results are displayed as relative changes compared to normoxic, untreated cells. In all types of cells treated with KB-R7943 (KBR) for 24 or 48 h in normoxic conditions, a slight acidification of intracellular pH occurred. Hypoxic treatment (1%; Hy) for 24 h decreased intracellular pH compared to normoxic control in all types of cells. After 48 h of hypoxic treatment, intracellular pH increased in all types of cells to a slightly alkaline pH. Under the same conditions, in the presence of KB-R7943 (HyKBR), intracellular pH remained acidic. Involvement of the NCX1 in intracellular acidification in hypoxia was verified using NCX1 siRNA (HysiNCX1) in SiHa cells, where rapid drop of the intracellular pH was observed after 48 h of hypoxia compared to scrambled siRNA (Hyscr; **D**). Results are displayed as mean ± S.E.M. and represent measurements of at least three independent cultivations. Statistical significance compared to normoxia *** corresponds to *p* <0.001, statistical significance compared to hypoxia + *p* < 0.05, ++ *p* < 0.01 and +++ *p* < 0.001.

**Figure 2 cancers-11-01139-f002:**
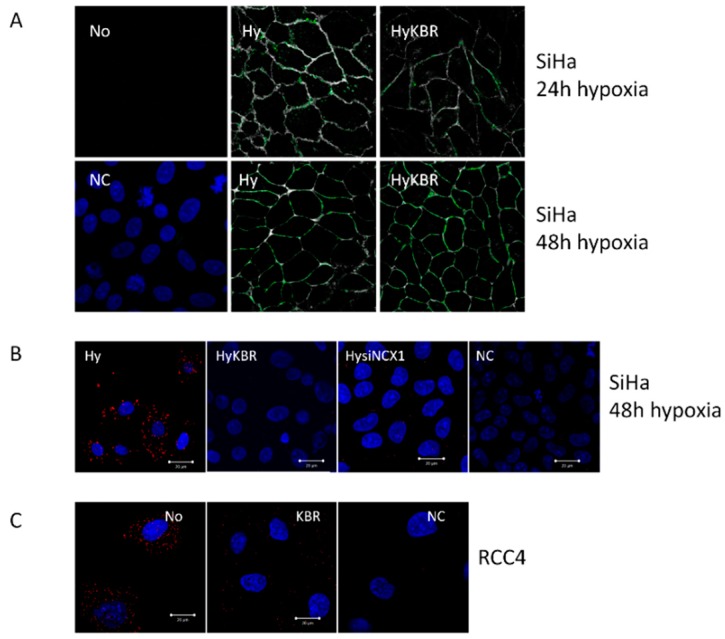
Co-localization of the NCX1 and CA IX. Co-localization of the NCX1 and CA IX in hypoxic SiHa cells (**A**). Cells were subjected to hypoxia (Hy) for 24 h (**A**, upper row) or 48 h (**A**, bottom row). Co-localization was determined by Co-localization Finder plugin, which was used to assess the co-localization of CA IX and NCX1 signals. The background pixels were excluded from the co-localization analysis, the co-localizing pixels were determined and marked in green (**A**). Co-localization of the NCX1 and CA IX in SiHa cells was detected also by proximity ligation assay after 48 h in hypoxia (Hy; **B**), where a clear red signal is visible in hypoxia. When NCX1 was silenced by NCX1 siRNA (HysiNCX1), PLA signal was much weaker, which corresponds to the effectivity of silencing (**B**). Also, PLA signal was not visible in SiHa cells treated by KB-R7943 (HyKBR) in the presence of hypoxia (**B**). In RCC4 cells that are spontaneously hypoxic, KB-R7943 (KBR) prevented co-localization of the NCX1 and CA IX (**C**). No—normoxia, NC—negative control.

**Figure 3 cancers-11-01139-f003:**
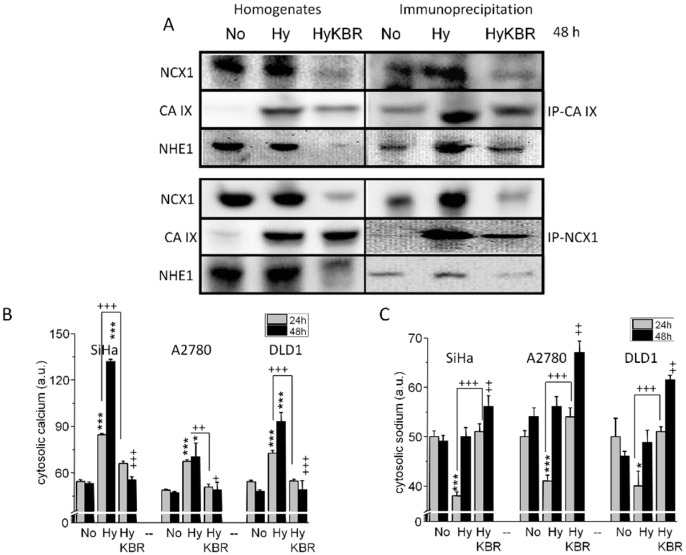
Immunoprecipitation with the NCX1 (**A**; IP-NCX1) or CA IX (**A**; IP-CA IX) antibodies. Homogenates from the SiHa cells (normoxic (No), hypoxic (Hy) and hypoxic treated with KB-R7943 (HyKBR) for 48 h) were immunoprecipitated with NCX1 antibody or CA IX antibody, blotted onto PVDF membrane and subsequently incubated with CA IX/NCX1 and NHE1 antibodies. Data show that NHE1 co-precipitates with NCX1 as well as with CA IX (**A**). In SiHa DLD1 and A2780 cells, cytosolic Ca^2+^ was increased by hypoxic stimuli after 24 (**B**, gray columns) and also 48 h (**B**, black columns), while in hypoxic cells treated with KB-R7943 (HyKBR), cytosolic calcium was comparable to control, normoxic cells (**B**, No). Cytosolic sodium level was decreased due to 24 h hypoxia (**C**) and this decrease was prevented in the hypoxic SiHa cells treated with KB-R7943 (HyKBR). After 48 h of hypoxia, cytosolic sodium levels were comparable with those from normoxic cells (**C**). Results (**B,C**) are expressed as mean ± S.E.M. and represent an average from at least three independent measurements. Statistical significance * represents *p* < 0.05 and *** *p* < 0.001 compared to normoxic controls. Statistical significance ++ represents *p* < 0.01 and +++ *p* < 0.001 compared to hypoxic untreated cells.

**Figure 4 cancers-11-01139-f004:**
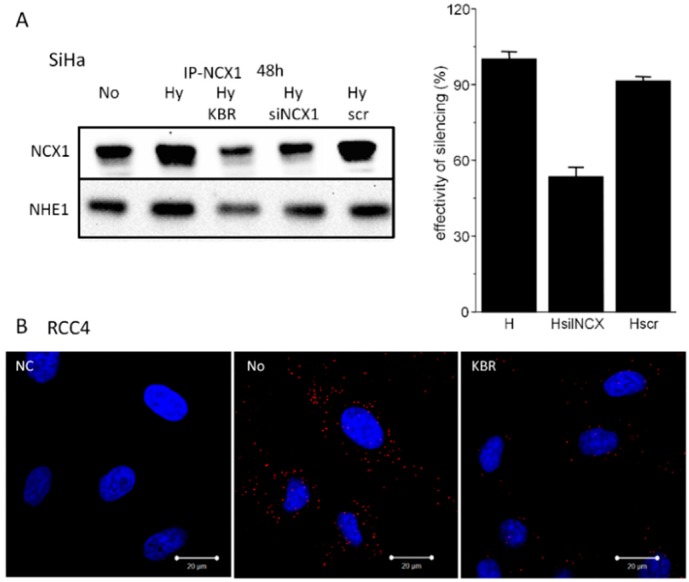
NCX1 immunoprecipitates with the NHE1, as determined by KB-R7943 (HyKBR), silencing of the NCX1 (HysiNCX1) in SiHa cells (**A**) and by proximity ligation assay in RCC4 cells (**B**). Effectivity of silencing of the NCX1 was approximately 50%, as determined by Western blot analysis. No—normoxia, NC—negative control, Hy—hypoxia for 48 h.

**Figure 5 cancers-11-01139-f005:**
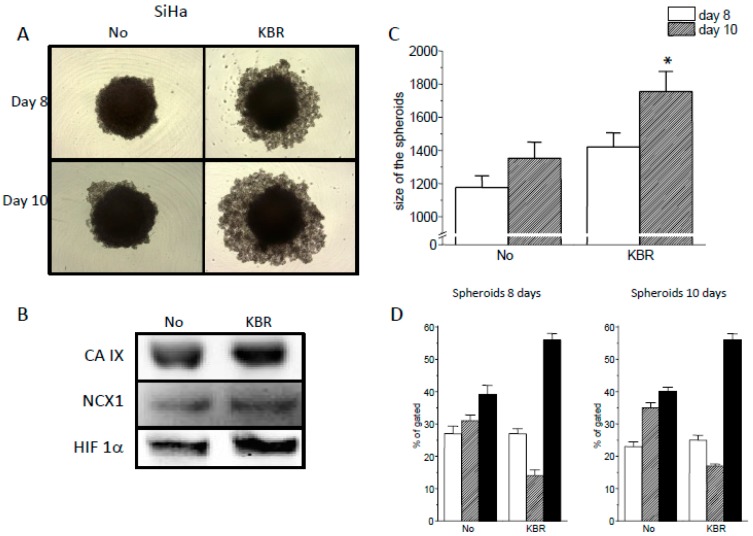
Effect of KB-R7943 on SiHa spheroids. SiHa spheroids were grown either in normal environment (No), or in the presence of KB-R7943 (KBR). Representative images of these spheroids at the day 8 and 10 are shown (**A**). After the day 10, spheroids were subjected to Western blot analysis, which proved the presence of CA IX, NCX1 and HIF-1α protein (**B**). In the presence of KB-R7943, SiHa spheroids disintegrated rapidly compared to untreated spheroids. Size of the spheroids at the day 8 and 10 is shown in the graph (**C**). Results are displayed as mean ± S.E.M., *n* = 15. Statistical significance *—represents *p* < 0.05 compared to control. (**C**). Number of apoptotic cells (**D**, striped columns) decreased in 8 and 10 days spheroids due to KB-R7943 treatment, while number of necrotic cells significantly increased (**D**, black columns). Empty columns represent number of alive cells in 8 and 10 days spheroids (**D**).

**Figure 6 cancers-11-01139-f006:**
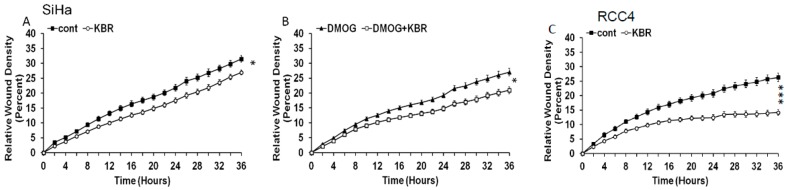
Effect of KB-R7943 on the SiHa and RCC4 migration. KB-R7943 decreased migration in normoxic SiHa cells (**A**), and decrease in KB-R7943 induced migration was even more pronounced when SiHa cells were theated with DMOG (**B**). In spontaneously hypoxic RCC4 cells, significant decrease in migration due to KB-R7943 treatment was also detected (**C**). Statistical significance * represents *p* < 0.05 and *** *p* < 0.001.

**Figure 7 cancers-11-01139-f007:**
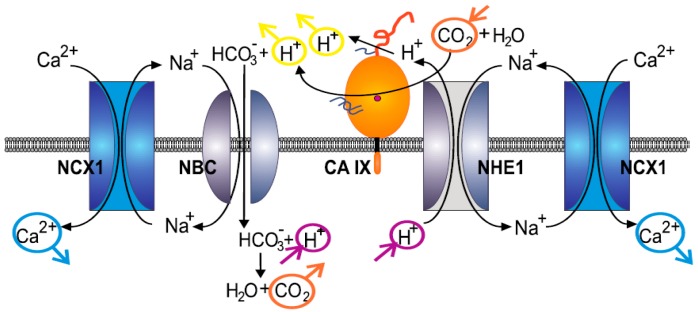
Schematic illustration of the proposed model of the cooperation between NCX1, CA IX, and NHE1. Right side: Based on the experiments presented here, hypoxia-induced CA IX can interact with both, NHE1 and NCX1. CA IX may facilitate proton export via NHE1 in exchange of Na^+^ import (presumably via a non-catalytic mechanism independent of CO_2_ hydration as explained by Jamali et al. [7]. Elevated intracellular Na^+^ then supports the reverse mode activity of NCX1 executing Na^+^ export coupled with Ca^2+^ import. Left side: Hydration of CO_2_ catalyzed by CA IX generates a surplus of extracellular protons in addition to HCO_3_^−^ ions. As known from previous studies, CA IX can interact with NBC1, which takes up the HCO_3_^−^ and performs its coupled import with Na^+^. Intracellular bicarbonate consumes intracellular protons producing CO_2_, which leaves the cell to pericellular space. As suggested above, NCX1 may operate in the reverse mode and execute Na^+^ efflux in exchange of Ca^2+^ influx. Thereby, this complex metabolon may contribute to intracellular alkalinization, extracellular acidification and intracellular accumulation of Ca^2+^ which is known to occur in certain types of cancers [34].

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
