# Peer review of "Type 1 Sodium Calcium Exchanger Forms a Complex with Carbonic Anhydrase IX and Via Reverse Mode Activity Contributes to pH Control in Hypoxic Tumors"

_cancers, 2019, doi:10.3390/cancers11081139_

Round 1
Reviewer 1 Report
Comments on Cancer MS 563034.
The authors described the role of NCX1 functioning in the reverse mode to pH control in hypoxic tumors.
Despite interesting datas, I am puzzled by how the experiments were done, especially the use of this NCX1 inhibitor and hypoxia.
For the reasons I will describe below, I recommend the authors to submit a new revised version.
Major concerns:
1. The main concern is the use of this inhibitor. The work done is almost based only on the use of KB-R7943. This inhibitor is used at 10 µM during 24-48h… A rapid search on PubMed shows that this inhibitor has also been described as a powerful inhibitor of TRP channels, which are expressed by a large number of cell types (Kraft 2007, BBRC). Several works have also described the mitochondria as a target of KB-R7943 (Wiczer 2015, BBRC; Santo-Domingo 2007, Br J Pharmacol) and logically mitochondria is impacted by hypoxia… Thus, it means that 24 to 48h KB treatment at 10 µM will trouble several targets affecting the global cell Ca2+ homeostasis, rendering the data analysis difficult. Despite these embarrassing papers, there is no discussion on KB-R7943 specificity.
Furthermore, by interfering with the cell homeostasis, KB-R7943 could have an impact alone on expression of various proteins. However, in fig 3, this control is missing: what about the expression of NCX1 in normoxia under KB-R7943 treatment ? What is the effect of KB-R7943 on cytosolic Ca2+ and Na+ concentrations in normoxia ?
2. NCX isoform. KB-R7943 is also an inhibitor of NCX2 and NCX3, are these NCX isorforms expressed by the cells used in this study ?
3. Hypoxia is not easy to get in laboratories. In the M&M, there is no description of how the cells are treated by hypoxia. Due to technical difficulties, it is now easier to treat cells with cobalt ions to mimick hypoxia and maintain these conditions. Could the authors give more details on how they get hypoxia ? What about the HIF1 transcription factor, that appeared in the end of the MS, a bit surprising for an hypoxia player.
4. reverse mode of NCX1. Except the KB-R7943 use with all the above described reserves, there is no clue in this work about NCX1 functioning in reverse mode. To favour the reverse mode, it is common to decrease the extracellular Na+ concentration, like done by Sun et al (2014), the reference number 23.
5. the number of cells used in the experiments ? In the M&M "statistical analysis", the authors wrote that each value represents an average of 3-6 wells. But how many cells per well ?
6. apoptosis/necrosis: in figure 5, an experiment was done to measure the number of apoptotic or necrotic cells. How to distinguish these two kind of death ?
Minor concerns:
1. Could the author explain why they maintain the cells in their culture medium (without serum) for intracellular Ca2+ and Na+ recordings ? Due to the complexity of these media, especially working with ions, it is more common to use very simple medium that can be easily modified.
About the Ca2+ recordings, did the authors use ionomycin to get the maximal fluo-3 fluorescence ? Without doing a calibration, it must be done to be sure that the fluo-3 loading is the same among the cells.
About the Na+ recordings, the author used ouabain, there is no explanation about the need for Na+-K+-ATPase inhibition. After the SBFI loading, the cells are washed by 5% glucose: in glucose-containing DMEM or really 5% glucose solution ? Next the recordings are made in which medium ?
After the recordings, it is difficult to understand how the results were expressed and showed in fig 3. How many cells were measured to get these histograms ?
2. siRNA. NCX1 silencing was made by the use of siRNA for 48h. The reduction only raised 50%. Next the experiments were done for 24 or 48h. Meaning 3 to 4 days after transfection. Did the authors verify the efficacy of the silencing after 3 to 4 days ?
3. Figure 4: the expression of CA IX is decreased by si NCX1 but also si scramble… any explanation ?
4. in the text values of decrease / increase are not done, only the significancy…
5. Line 208: "DMOG". What is it ?
Line 61: "under physiological conditions, NHE1 is essentially inactive". Reference ?
Line 206: Since pH regulation IS instead in
Reviewer 2 Report
Manuscript ID cancers-563034 by Liskova V. et al. describes an interesting study about the role of NCX1 in the regulation of extracellular pH of hypoxic tumors. Moreover, authors suppose that NCX1 effects are due to the formation of a NCX1/CA IX/NHE1 metabolon.
Although the work is quite clear, I suggest to ameliorate both the introduction section, pointing out the potential importance both of CA IX inhibitors in tumor progression (lines 55-56 pag. 2) as well as of iNOS inhibitors (lines 78-79 pag. 2), and the discussion section analyzing in more details the obtained results.
Finally, I recommend to revise language and style, as I found different typos and mistakes.
Among the others, I would point out that the bicarbonate ion is HCO3- (number 3 in subscript and - in apex) and not HCO3- as reported (line 243 pag 9)
Author Response
REVIEWER 2
Manuscript ID cancers-563034 by Liskova V. et al. describes an interesting study about the role of NCX1 in the regulation of extracellular pH of hypoxic tumors. Moreover, authors suppose that NCX1 effects are due to the formation of a NCX1/CA IX/NHE1 metabolon.
Although the work is quite clear, I suggest to ameliorate both the introduction section, pointing out the potential importance both of CA IX inhibitors in tumor progression (lines 55-56 pag. 2) as well as of iNOS inhibitors (lines 78-79 pag. 2), and the discussion section analyzing in more details the obtained results.
We ameliorated both the introduction and discussion sections and added the information as suggested by the reviewer.
Finally, I recommend to revise language and style, as I found different typos and mistakes.
Manuscript was revised according to the reviewer’s suggestion.
Among the others, I would point out that the bicarbonate ion is HCO3- (number 3 in subscript and - in apex) and not HCO3- as reported (line 243 pag 9)
We apologize for this mistyping, it was corrected.
Reviewer 3 Report
In the manuscript submitted by Liskova et al., authors employing different approaches, analyze in vitro the role of the sodium calcium exchanger (NCX1) in pH modulation in hypoxic tumors (renal, ovarian and colon carcinomas). The results obtained suggest that NCX1 could coordinate a complex with carbonic anhydrase IX (CAIX) and sodium proton exchanger type 1 (NHE1) contributing to pH regulation in hypoxic tumors.
However, the authors might want to consider these major points:
- In my opinion the Figures must be improved. For instance, in Figure 1 bars and error bars are misaligned. In Figures 2 and 4B scale bars are hard to see. The resolution of the bands in Figure 3A and 4A are very low, whilst Figure 3B and C present, again, bar and error bars misaligned. In Figure 5A representative images of these spheroids at the day 15 is missing, as contrarily reported in the figure legend. In Figure 6D the authors affirm: “In spontaneously hypoxic RCC4 cells, significant decrease in migration due to KB-R7943 treatment was also detected”. Frankly, I can not appreciate the “decrease” in these pictures.
- All the text in figure legends should be justified
- Line 103: The statistical significance described is quite difficult to follow
- Line 117: …. “what results in subsequent acidification of the extracellular pH (not shown)”. Why do not show this data ?
- Line 200: I can not understand the significance of the “change in the size of organoids”
- Lines 251-258 and “Conclusions”: Please, rewrite these sentences
- Figure 7 should be moved at the end of results
Minor points:
- Please use symbols. For instance, Ca2+ instead of “calcium”. The same for sodium, bicarbonate, hours….
Author Response
REVIEWER 3
In the manuscript submitted by Liskova et al., authors employing different approaches, analyze in vitro the role of the sodium calcium exchanger (NCX1) in pH modulation in hypoxic tumors (renal, ovarian and colon carcinomas). The results obtained suggest that NCX1 could coordinate a complex with carbonic anhydrase IX (CAIX) and sodium proton exchanger type 1 (NHE1) contributing to pH regulation in hypoxic tumors.
However, the authors might want to consider these major points:
- In my opinion the Figures must be improved. For instance, in Figure 1 bars and error bars are misaligned. In Figures 2 and 4B scale bars are hard to see. The resolution of the bands in Figure 3A and 4A are very low, whilst Figure 3B and C present, again, bar and error bars misaligned. In Figure 5A representative images of these spheroids at the day 15 is missing, as contrarily reported in the figure legend. In Figure 6D the authors affirm: “In spontaneously hypoxic RCC4 cells, significant decrease in migration due to KB-R7943 treatment was also detected”. Frankly, I can not appreciate the “decrease” in these pictures.
We went through all Figure and redraw some of them (i.e. Figure 4A, Figure 5C, Figure 5D), although in our computer, the resolution meets the standard of this journal. Nevertheless, when we opened the same file in different type of computer and other software, the figures were disarranged. Therefore, we converted each figure into pdf file and copied these files into the manuscript. We are grateful for this comment and we will ask editorial office to check for the quality of figures.
In figure 5 we overlooked the incorrect figure legend. We studied spheroids in days 8 and 10 and quantificantion was performed from at least 15 spheroids. We corrected the figure legend and we apologize for this mistake.
In Figure 6D, were obtained from Incucyte ZOOM kinetic imaging system. Since we were not able to improve images of scratches, we remove these photos from the Figure.
- All the text in figure legends should be justified
Figure legends were justified.
- Line 103: The statistical significance described is quite difficult to follow
We described the statistics in more detail as follows:
Statistical significance compared to normoxia - *** represents p‹0.001, statistical significance compared to hypoxia - + - p ‹0.05, ++ - p‹0.01 and +++ -p‹0.001.
- Line 117: …. “what results in subsequent acidification of the extracellular pH (not shown)”. Why do not show this data ?
While the intracellular pH was determined precisely by the fluorescent dye BCECF-AM, extracellular pH was determined just by pH electrode. We always observed decrease in the extracellular pH after 48 hours, but sometimes to different extent. Originally, we have these results in the graph, but finally we decided that more sophisticated approach will be necessary to quantify these results. We decided to remove the note about the extracellular pH from the manuscript.
- Line 200: I can not understand the significance of the “change in the size of organoids”
Size of the untreated and KB-R7943 treated spheroids was determined after 8 and 10 days. For each group, at least 15 spheroids were measured and from these results the graph was drawn. We apologize for missing statistics in the graph and not correct description in the figure legend; we corrected it in the revised version.
- Lines 251-258 and “Conclusions”: Please, rewrite these sentences
Both above-mentioned parts were rewritten as follows:
Lines 251-258: Based on our results, we hypothesized about the physiological relevance of this complex. Since both, CA IX and NHE1 are activated in hypoxic conditions and participate in proton extrusion, proton transport from cells could be considered as an adaptive mechanism in hypoxia that is essential for the cell’s survival [37,39]. Although function of these proteins in hypoxic conditions is known, their mutual co-operation and co-localization with NCX1 was not described until now. We have shown that KB-R943 as a blocker of the reverse mode NCX reduced migration rate of RCC4 cells. Role of the NCX in migration was already described e.g. in human gastric myofibroblasts [40] or microglia [41]. Complex formation probably increases extracellular pH in a microdomain, which might boost migration specifically from that region.
Conclusions: In summary, in the present study we show that the reverse-mode NCX1 forms a membrane complex with the CA IX and NHE1. This membrane complex not only represents a potent modulator of intracellular/extracellular pH, but may also play a role in tumor cell´s migration (and metastasis formation). Since NCX1 forms complex also with the β1- and β3- but not β2-adrenoceptors [44], it is likely that this metabolon is composed of additional protein components that can further modulate cellular responses to pH imbalances in tumor microenvironment.
- Figure 7 should be moved at the end of results
Figure was moved, as requested by this reviewer. From the spatial reasons it was placed at the beginning of Discussion.
Minor points:
- Please use symbols. For instance, Ca2+ instead of “calcium”. The same for sodium, bicarbonate, hours….
Manuscript was checked and words were replaced by corresponding symbols.
Round 2
Reviewer 1 Report
The authors provided responses to almost all my concerns.
Reviewer 3 Report
The authors have satisfied my concerns; the manuscript is now acceptable for publication